# Improving the Protectiveness of 3-Mercaptopropyl-Trimethoxysilane Coatings on Bronze by Addition of Oxidic Nano- and Microparticles

**Cecilia Monticelli** [1,*] **, Federica Zanotto** [1] **, Vincenzo Grassi** [2] **, Mahla Seyedi** [1] **and Andrea Balbo** [1]

1   "A. Daccò" Corrosion and Metallurgy Study Centre, Engineering Department, University of Ferrara, 44122 Ferrara, Italy; federica.zanotto@unife.it (F.Z.); mahla.seyedi@unife.it (M.S.); andrea.balbo@unife.it (A.B.)
2   "A. Daccò" Corrosion and Metallurgy Study Centre, University of Ferrara, 44122 Ferrara, Italy; vincenzo.grassi@unife.it
*   Correspondence: cecilia.monticelli@unife.it

**Abstract:** The protectiveness of coatings based on 3-mercaptopropyl-trimethoxysilane (PropS-SH) applied on bronze artworks was enhanced by investigating and tuning a variety of factors. In particular, the coating performances were optimized by proper additive choice among oxide nanoparticles ($CeO_2$, $La_2O_3$, $TiO_2$) and microparticles (fly ash (FA)), and by varying the aging time of the nanoparticle suspensions before coating application and the RT curing time of the coatings. Moreover, the possibility of conservation of the silane solution under refrigerated conditions was assessed. The aggressive environment was a tenfold concentrated synthetic acid rain (AR × 10). The techniques adopted comprised electrochemical tests (polarization curve recording and EIS tests) and SEM-Energy Dispersion Spectroscopy (EDS) observations. In the case of FA microparticles, never used before as additives in silane coatings, further tests were performed in FA suspensions in AR × 10 to clarify the contribution of these particles to PropS-SH coating protectiveness. The tests included pH measurements, elemental chemical analyses, and electrochemical tests on bare bronze. Improved performances of PropS-SH coatings were achieved by $La_2O_3$, $CeO_2$, and FA addition, with $La_2O_3$ affording the best results during 20 days of immersion. The positive influence of FA was connected to its alkaline character and to the release of soluble silicates.

**Keywords:** organosilane coatings; synthetic acid rain; oxide nanoparticles; fly ash; bronze

## 1. Introduction

Because of its many applications in architecture for structural and artistic elements, bronze often operates in continuous or cyclic contact with rain for long periods (even for centuries in the case of sculptures and other decorative artworks). These exposure conditions cause various corrosion forms, such as stains, stripes, or crusts, that compromise the overall surface aspect—and with time, even the artwork mechanical properties [1,2]. Therefore, it is of utmost importance the development of nontoxic protective coatings, which in the cultural heritage field must be characterized by natural aspect and reversibility or at least easy surface retreatability, allowing for multiple restoration treatments. The commonly adopted treatments include cleaning and then application of commercial products, such as Incralac, which contains a thermoplastic methyl methacrylate copolymer dissolved in rather toxic solvents (e.g., toluene and xylene) and admixed with benzotriazole (used as corrosion inhibitor and anti-UV agent) [3].

Recently, promising results were also achieved with organosilane formulations [4–6], which proved to be less toxic than Incralac on the basis of standard occupational hazard tests [7]. In all cases, the coatings were produced by previous hydrolysis of the silicon alkoxide bonds and subsequent condensation of the obtained silanol groups inside the coatings to produce the siloxane network. High temperature curing, in the range 100–200 °C, is usually ensured in the industrial field to promote the condensation reaction [8], but it cannot be applied to cultural heritage artworks for which room temperature (RT) curing is generally prescribed.

Many literature studies deal with silane coatings for industrial applications where they operate as both primers/adhesion promoters and single protective treatments for various metals and alloys [9,10]. In some cases, their protectiveness was increased by the addition of oxide nanoparticles, such as $CeO_2$ (applied for the protection of galvanized steel [11–14], aluminium alloys [15,16], steel [17], and Mg alloys [18]), $SiO_2$ [19–23], $La_2O_3$ [24], $TiO_2$ [25], and $Y_2O_3$ [26]. However, other nanoparticles too were tested, such as clay [27,28], $CeO_2 \cdot ZrO_2$ [29], $CeAlO_3$ [30], and layered double hydroxides [31]. In general, these additives positively affected the coating performances by inducing higher coating thicknesses [20,21,23,30] or by partially filling the pores of the silane film, so increasing the silane film density [24]. Some of them acted as cathodic inhibitors [18,20,26,31]. $CeO_2$ nanoparticles were reported to activate other specific mechanisms for increasing the coating protectiveness, that is formation of Ce–O–Si bonds and consequently production of denser coatings [15], acceleration of Si-OH group condensation, thus increasing the silane film thickness [17] and easy incorporation into the nanoparticle structure of aggressive anionic species, like chlorides, by forming charge-compensating defects in the oxygen sub-lattice [12].

Difficulties in the development of barrier coatings derived from nanoparticle agglomeration or addition of excessive nanoparticle amounts that can accelerate the coating delamination and facilitate the water uptake through the pores of the surface film [22,23].

To our knowledge, only two papers dealt with the effectiveness of particle-charged silane coatings on copper and copper alloys. In the first one, the effects of a 13-month exposure to a marine urban atmosphere of 3-mercaptopropyl-trimethoxysilane (PropS-SH) coatings applied on bronze were described. A reduced tendency to UV/thermal degradation was detected for coatings charged by $CeO_2$ nanoparticles, micrometric $La_2O_3$ particles and, particularly, $TiO_2$ nanoparticles. Differential Scanning Calorimetry (DSC) and Fourier Transform Infrared (FTIR) spectroscopy tests suggested that the particles improved the coating performance mainly by assisting silane reticulation. $TiO_2$ nanoparticles were the most effective, likely because they exerted a specific action as UV light absorber [32]. In another paper, Fan et al. [24] studied the influence of $La_2O_3$ nanoparticles on the barrier properties of PropS-SH coatings against brass corrosion in NaCl solution. $La_2O_3$ nanoparticles increased the rate of silane condensation reaction and effectively filled the defects of the silane film; thus, reducing the anodic reaction of brass corrosion.

This paper aims at formulating a protocol for optimizing the protectiveness of PropS-SH coatings against bronze corrosion in concentrated synthetic acid rain. The analyzed parameters of coating preparation include first of all the shelf life of the hydrolyzed silane solution at 4 °C. Actually, too long storage times can produce undesirable condensation phenomena inside the silane solution [33]. Then, the influence of different types of oxidic nanoparticles ($CeO_2$, $La_2O_3$, $TiO_2$) and microparticles (Class F fly ash) in the silane formulation was assessed, including the choice of a sufficient aging time of nanoparticle suspensions in silane solution before coating application. Finally, the possibility of an efficient RT curing for the coatings before their exposure to the aggressive environment was assessed.

## 2. Materials and Methods

### 2.1. Materials

The specimens were cut from bronze ingots obtained by casting in stone molds. They had composition similar to those of Renaissance artefacts (91.9 Cu, 2.9 Zn, 2.4 Sn, 1.0 Pb, 0.8 Sb wt.%) and

were characterized by a dendritic microstructure of cored alpha solid solution, with Sn-rich precipitates and Pb globules in the interdendritic spaces. Some rather uniformly-distributed porosity was also present [34].

Bronze electrodes with exposed surface area of about 1 cm$^2$ were embedded in epoxy. They were ground by emery paper, polished down to 1 μm roughness, washed with deionized water, and finally degreased with acetone.

The protective coatings were based on 3-mercaptopropyl-trimethoxysilane (PropS-SH, purity 95%, Aldrich cod. 175617). Previous tests carried out in our laboratory [35,36] addressed the influence of composition and pH of PropS-SH solution on the coating protectiveness. Based on these investigations, a constant practice was adopted in coating preparation:

- the basic silane formulation consisted of a hydro-alcoholic solution with composition 90 vol.%/5 vol.%/5 vol.% ethanol/water/silane;
- pH of the silane was set at 4 by adding some drops of diluted sulfuric acid solution;
- the coatings were prepared by dip coating, that is, by immersing the bronze electrodes in the hydro-alcoholic silane solutions for 1 h, followed by fast withdrawal;
- a RT curing was adopted to comply with cultural heritage requirements.

Given these constant conditions, some parameters were modified to assess the silane solution shelf life and to improve the coating performances. The variable parameters were:

- the storage time of the silane solution at 4 °C. Values of 7 days, 30 days or 1 year were tested. The subsequent coating preparation was always performed at RT;
- the addition of 250 ppm nano- or microparticles. The nanoparticles (CeO$_2$ (nano-sized, Aldrich cod. 544841), La$_2$O$_3$ (nano-sized, Aldrich cod. 634271), and TiO$_2$ (nano-sized, Aldrich cod. 718467)) were dispersed into the hydrolyzed silane solution by 15 min ultrasonication. Moreover, the effect of micrometric Class F Fly Ash (FA) addition in the coatings was investigated. FA is a waste product of thermal power plant (from Aboño, Asturias, Spain), which was used after sieving to afford a maximum particle size of 30 μm (Figure 1). Its composition, reported in Table 1, reveals it is a silica- and alumina-rich powder, which, according to X-Ray Diffraction, contains crystalline mullite, quarz, and magnetite, and large amounts of amorphous phases;
- the aging time of nanoparticle suspensions in silane solution, before coating application. The coating performances were evaluated in the absence and in the presence of a 7-day aging of the nanopowder suspensions at 4 °C;
- the coating curing time at RT before immersion in the aggressive solution. Values of 1 or 10 days were tested.

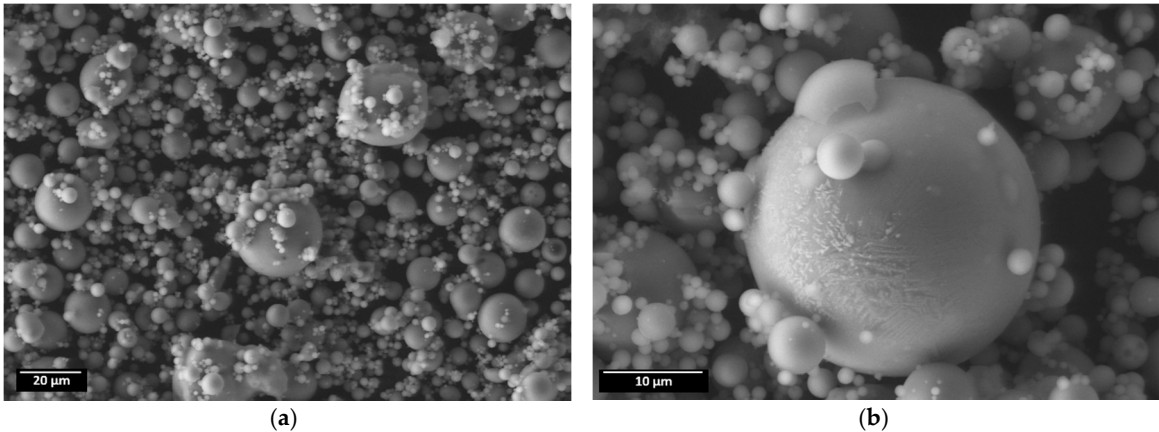

| (**a**) | (**b**) |

**Figure 1.** Micrographs of FA powder obtained by scanning electron microscope at lower (**a**) and higher (**b**) magnification.

**Table 1.** Main oxide composition of fly ash (FA) powder.

| Material | L.o.I.[1] | IR[2] | $SiO_2$ | $Al_2O_3$ | $Fe_2O_3$ | CaO | MgO | $SO_3$ | $K_2O$ | $Na_2O$ | MnO | $TiO_2$ |
|---|---|---|---|---|---|---|---|---|---|---|---|---|
| Fly Ash | 3.59 | 2.23 | 51.78 | 27.80 | 6.18 | 4.59 | 1.52 | 0.71 | 2.51 | 0.59 | 0.06 | 1.35 |

[1] Loss on Ignition, [2] Insoluble Residue.

The aggressive solution was a tenfold concentrated synthetic acid rain (AR × 10) with pH 3.3, with composition and electrical conductivity, as reported in Table 2 [37]. In few cases, the tests were performed in 3.5% NaCl. The aggressive solutions were thermostated at 30 °C.

**Table 2.** Characteristics and composition of the tenfold concentrated synthetic acid rain (AR × 10).

| Species | Concentration/mg·L$^{-1}$ |
|---|---|
| $Cl^-$ | 12.7 |
| $NO_3^-$ | 46.4 |
| $NH_4^+$ | 10.6 |
| $SO_4^{2-}$ | 19.4 |
| $HCOO^-$ | 0.5 |
| $CH_3COO^-$ | 2.3 |
| $Na^+$ | 5.3 |
| $Ca^{2+}$ | 3.4 |
| **Electrical Conductivity/μS·cm$^{-1}$ (at 25 °C)** | 345 |
| **pH** | 3.3 |

## 2.2. Corrosion Tests and Characterization Methodology

For electrochemical tests, a three-electrode configuration cell was adopted, including a Pt electrode as a counter and a Saturated Calomel Electrode (SCE) as a reference. All the potentials in the text are referred to SCE.

Electrochemical Impedance Spectroscopy (EIS) was adopted for corrosion behavior monitoring during 20 days of immersion, under the following experimental conditions: ±10 mV rms alternating potential signal; 25 kHz–1 mHz frequency range; 5 frequencies/decade. The apparatus used for these tests was a Frequency Response Analyzer (FRA) 1260 Solartron–Schlumberger coupled to an EG&G/PAR Model 273A potentiostat/galvanostat (Ametek, Berwyn, PA, USA). In order to investigate the corrosion mechanism, the EIS spectra were fitted to proper equivalent circuits (ECs) by ZView Software (v. 3.5g, Scribner Associates Inc., Southern Pines, NC, USA). The tests were performed in triplicate.

Anodic and cathodic polarization curves were recorded at the end of the 20-day immersion period, starting from the corrosion potential ($E_{cor}$). The potential scanning rate was 0.166 mV/s.

## 2.3. Surface Analyses

Surface analyses were carried out on bare and coated bronze specimens after 20 days of immersion under free corrosion conditions. The tests were performed by a Scanning Electron Microscope (SEM model ZEISS EVO MA15, Oberkochen, Germany), operating under an acceleration voltage of 20 KV and equipped by an Energy Dispersion Spectroscopy (EDS) system (X-ACT Oxford Instruments, Oxford, UK). SEM-EDS permitted to characterize the surface morphology and the extent of the corrosion attack on bronze substrate.

## 2.4. Tests in FA Suspensions

Further tests were carried out to clarify the action of FA in the coatings. They consisted in immersions of uncoated bronze electrodes in AR × 10 in the absence and in the presence of 1 g/liter FA powder (kept under mild stirring). Before immersion, the FA suspension was aged for 24 h at 30 °C

and pH values were monitored during both the aging and the subsequent 20-day immersion period. The EIS tests, carried out throughout the immersion period to monitor bronze corrosion behavior, and the final polarization curves were performed as described in Section 2.2.

The concentrations of Al, Si, Ca, Fe, and Mg elements leaching from FA particles into AR × 10, during the initial 24 h suspension aging, were evaluated after suspension filtering by Inductively Coupled Plasma-Optical Emission Spectroscopy (ICP-OES, Agilent, Santa Clara, CA, USA), while the concentrations of Na and K in the same solution were detected by Atomic Absorption Spectroscopy (AAS, Perkin-Elmer, Waltham, MA, USA).

## 3. Results

### 3.1. Influence of Silane Solution Storage Time

Some preliminary tests aimed at evaluating the effects of a long storage time of the hydroalcoholic silane solution at 4 °C on the final coating protectiveness. The polarization curves in Figure 2 show the corrosion behavior of bronze electrodes in the presence of coatings produced after solution storage times of 7, 30, or 365 days (other test conditions: 1-day curing at RT, 7-day immersions in AR × 10). The Figure shows that an increase in storage time up to 1 year induced a continuously improved coating protectiveness. After a 7-day storage, only the cathodic reaction of oxygen reduction was slightly hindered, while the anodic curve partly overlapped that recorded on uncoated bronze. After storage times of 30 days and 1 year, the cathodic and anodic curves progressively shifted to lower currents. This means that at 4 °C silane hydrolysis in the hydroalcoholic solution slowly proceeded, producing more silanol groups available for coating reticulation, so that an improvement in the coating barrier properties against the aggressive environment was achieved. This also indicates that until 1 year, the condensation degree of silanol groups within the solution was negligible, because its occurrence would have prevented the correct coating buildup [33]. The importance of a correct silane hydrolysis in the development of good coating properties was already stressed by other researchers [38]. The results here described suggest that the silane formulation can be stored at 4 °C for at least 1 year.

All tests described from here on were performed on coatings prepared after about 30 days of hydrolysis at 4 °C.

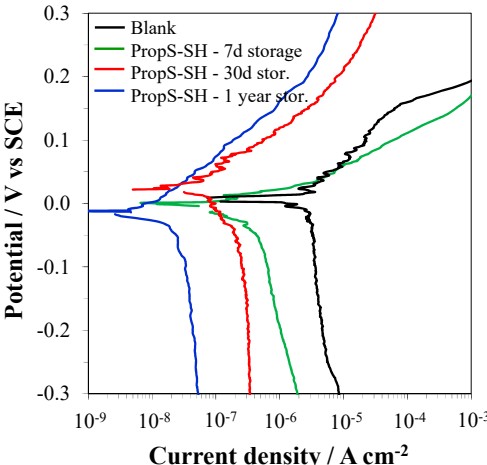

**Figure 2.** Polarization curves recorded after 7 days of immersion in AR × 10 at 30 °C, on bare bronze and 3-mercaptopropyl-trimethoxysilane (PropS-SH) coated bronze. The coatings were obtained after different storage times of the silane solution at 4 °C. Curing of 1 day at RT.

### 3.2. Influence of RT Curing Time

Prolonging the RT curing time of the coatings from 1 to 10 days determined a significant improvement in the coating properties. In fact, the polarization curves collected in Figure 3 show that

after 1-h immersion, the bronze electrodes exhibited decreasing corrosion currents ($i_{cor}$) from about 0.07 µA cm$^{-2}$ (1-day cured coating), to 0.011 µA cm$^{-2}$ (10-day cured coating), with a continuous shift of both cathodic and anodic curves to lower currents. This was ascribed to the achievement of a higher coating reticulation degree before exposure to the aggressive solution [34]. A curing time of 10 days was adopted in the following tests.

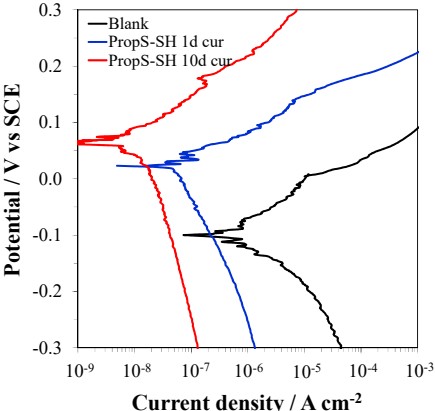

**Figure 3.** Polarization curves recorded after 1 h of immersion in AR × 10 at 30 °C, on bare bronze and PropS-SH coated bronze. The coatings were obtained after 30 days of storage at 4 °C and were exposed to the aggressive environment after different RT curing times.

The long time behavior of coated bronze electrodes prepared under these conditions is shown in Figure 4. In particular, the Figure collects the polarization curves recorded at different immersion times in AR × 10 on bare and PropS-SH coated bronze. Concerning bare bronze, after 1-h immersion the alloy was under passive conditions (or pseudopassive conditions, taking into consideration the not negligible values of the passive currents, in the range 1–10 µA·cm$^{-2}$) with corrosion current ($i_{cor}$) of about 1.6 µA cm$^{-2}$ and breakdown potential of about 0.01 V(SCE). At this short immersion time, the cathodic reaction of oxygen reduction on bronze was under activation control and rather slow, and then it accelerated after 1 day of immersion, inducing a fast $E_{cor}$ ennoblement to the breakdown potential and an $i_{cor}$ stimulation. After 20 days, the cathodic polarization curve remained close to that obtained at 1 day of immersion, but the anodic currents were lower, likely due to the accumulation of surface corrosion products [39]. Table 3 collects the $E_{cor}$ and $i_{cor}$ values measured during these tests.

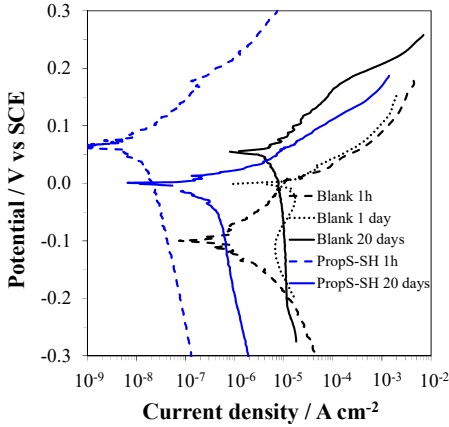

**Figure 4.** Polarization curves recorded on bare bronze and PropS-SH coated bronze after different immersion times in AR × 10 at 30 °C.

The application of PropS-SH coatings induced an excellent inhibition of both cathodic and anodic reactions after 1 h of immersion, with $i_{cor}$ of 0.015 µA·cm$^{-2}$. However, at the end of the 20-day

immersion, PropS-SH only hindered the cathodic reaction of oxygen reduction, so allowing final $i_{cor}$ values of 0.5 μA·cm$^{-2}$ (Table 3). According to Chen et al. [38], the long Si–O–Si chains may break down during immersion, so generating a number of film defects.

**Table 3.** Corrosion potentials ($E_{cor}$) and corrosion current densities ($i_{cor}$) obtained on bare and PropS-SH coated bronze electrodes in AR × 10 at 30 °C.

| Coating | Immersion Time | $E_{cor}$/V(SCE) | $i_{cor}$/μA·cm$^{-2}$ |
|---------|----------------|------------------|------------------------|
| Blank | 1 h | −0.100 | 1.6 |
| | 20 days | 0.055 | 8.3 |
| PropS-SH | 1 h | 0.066 | 0.015 |
| | 20 days | 0.001 | 0.5 |

### 3.3. Influence of Nano and Microparticle Addition in Coating Formulations

Nano- and microparticle addition was intended to improve the coating barrier properties at long immersion times. Unfortunately, in the absence of any aging time of the nanoparticle suspensions before coating preparation, the CeO$_2$–, La$_2$O$_3$–, or TiO$_2$–charged coatings did not exhibit any improved protectiveness in comparison to plain PropS-SH. This is shown in Figure 5a collecting the polarization curves recorded after 20 days of immersion in AR × 10 and in Table 4 where the related $E_{cor}$ and $i_{cor}$ values are collected. The Figure evidences that La$_2$O$_3$ determined an increase in the anodic and cathodic overvoltages but no meaningful decrease in i$_{cor}$, while CeO$_2$ and TiO$_2$ stimulated the corrosion process (Table 4). On the contrary, coatings charged by FA microparticles exhibited much lower anodic and cathodic currents with final $i_{cor}$ of 0.043 μA·cm$^{-2}$, instead of 0.5 μA·cm$^{-2}$ in plain PropS-SH (Figure 5b, Tables 3 and 4).

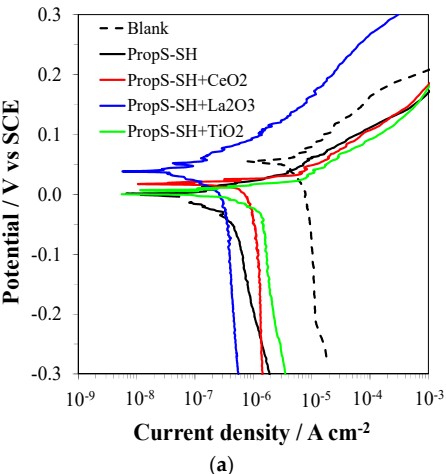
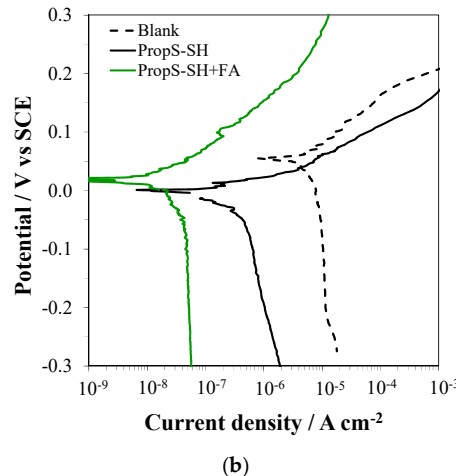

(a)        (b)

**Figure 5.** Polarization curves recorded after 20 days of immersion in AR × 10 at 30 °C, on bare (blank) and coated bronze. The coatings include plain PropS-SH and PropS-SH containing CeO$_2$, La$_2$O$_3$, or TiO$_2$ nanoparticles (**a**) and FA microparticles (**b**). Conditions of coating preparation: 30-day storage time of the silane solution at 4 °C, 10 day curing at RT.

In the presence of CeO$_2$ and La$_2$O$_3$, an improvement was achieved by allowing a 7-day aging of the nanoparticle suspensions at 4 °C before the RT coating preparation. Under these conditions of coating preparation, the polarization curves recorded after 20 days of immersion in AR × 10 are displayed in Figure 6. CeO$_2$ and La$_2$O$_3$ additions actually induced more protective coatings, as documented by the lower cathodic and anodic currents, in comparison to those recorded on plain PropS-SH. The $i_{cor}$ values of bronze diminished as well (Table 4). In the case of TiO$_2$–charged coating, the preliminary aging did not improve the coating protectiveness.

**Table 4.** Corrosion potentials ($E_{cor}$) and corrosion current densities ($i_{cor}$) obtained on particle-containing PropS-SH coatings after 20 days of immersion in AR × 10 at 30 °C. η corresponds to the increase in protection efficiency afforded by particle charging.

| Coating | Aging Time/days | $E_{cor}$/V(SCE) | $i_{cor}$/µA·cm$^{-2}$ |
|---|---|---|---|
| PropS-SH + CeO$_2$ | 0 | 0.017 | 1.1 |
| PropS-SH + La$_2$O$_3$ | 0 | 0.038 | 0.31 |
| PropS-SH + TiO$_2$ | 0 | 0.004 | 1.5 |
| PropS-SH + FA | 0 | 0.018 | 0.041 |
| PropS-SH + CeO$_2$ | 7 | 0.030 | 0.08 |
| PropS-SH + La$_2$O$_3$ | 7 | 0.042 | 0.045 |
| PropS-SH + TiO$_2$ | 7 | −0.015 | 0.40 |

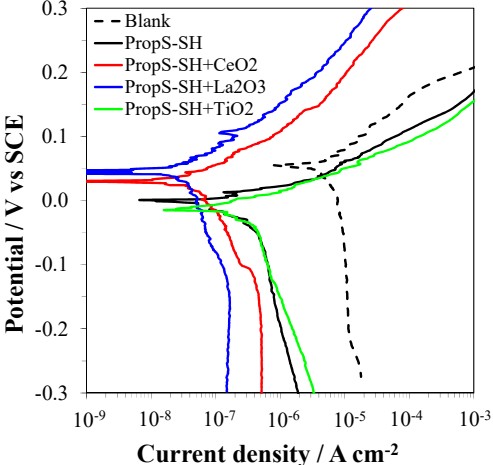

**Figure 6.** Polarization curves recorded after 20 days of immersion in AR × 10 at 30 °C, on bare (blank) and coated bronze. The coatings consist of plain PropS-SH and PropS-SH charged by CeO$_2$, La$_2$O$_3$, or TiO$_2$ nanoparticles. Conditions of coating preparation: 30-day storage time of the silane solution at 4 °C, 7-day aging of the suspensions at 4 °C, 10-day curing at RT.

## 3.4. EIS Analysis of PropS-SH Coating Protectiveness in the Presence of Nano and Microparticles

The EIS technique was adopted to investigate the corrosion behavior of bronze electrodes coated by PropS-SH in the absence and in the presence of the most effective particles, i.e., CeO$_2$ and La$_2$O$_3$ nanoparticles and FA microparticles, during immersion in AR × 10 solutions. EIS spectra on bare bronze were analyzed in a previous paper [39], and the results are here reported as a reference.

Figure 7 shows that after 20 days of immersion, and in general at immersion times longer than 1 h [39], bare bronze (blank) exhibited EIS spectra characterized by three capacitive time constants (τ) in the following frequency ranges (black arrows in Figure 7): 10–10$^3$ Hz (high-frequency (HF) τ), 10$^{-1}$–10 Hz (at slightly higher frequencies for 20-day immersion; medium-frequency (MF) τ), and at frequencies lower than 10$^{-1}$ Hz (low-frequency (LF) τ). Therefore, the EIS spectra were fitted by an equivalent circuit (EC) consisting of three nested R-CPE (Resistance-Constant Phase Element) couples, including [40,41]:

- a R$_f$-CPE$_f$ couple, related to the dielectric properties of the surface corrosion product film,
- a R$_t$-CPE$_{dl}$ couple, correlated to the charge transfer reaction, and
- a R$_F$-CPE$_F$ couple, due to faradaic reactions in the patina layer.

CPE elements were used instead of capacitances to compensate for surface roughness, porosities and other surface inhomogeneities. The impedance expression of the CPE was $Z = [Y(jw)^n]^{-1}$, where $w = 2\pi f$ is the angular frequency, $j = \sqrt{(-1)}$ is the imaginary unit, Y is a frequency independent value and $n$ is a fit parameter with values $0 \leq n \leq 1$, which measure the element deviation from the ideal

capacitive behavior (exhibiting $n = 1$) [42–44]. The CPE elements were converted into capacitances by the general equation $C = (R^{1-n} Y)^{1/n}$ (where Y is the parameter of each CPE element and $R$ the resistance in the corresponding $R$-CPE couple) [41,45]. After 1-h immersion, the alloy exhibited a passive (or pseudopassive) behavior but the thin passive film, although capable to slow down the charge transfer process, did not determine any HF loop. Therefore, the EC used to fit the spectra at this short immersion time did not include the $R_f$-CPE$_f$ couple.

Even in the presence of the PropS-SH coating, at immersion times of 1 day or longer, three capacitive $\tau$ were detected (blue arrows in Figure 7b), but only the MF one had the same meaning as in the blank. In fact, in agreement with previous findings [38,46], they were attributed respectively to the dielectric properties of the silane layer (the HF $\tau$ at frequencies $3 \times 10^4$–10 Hz) and to processes occurring sequentially through the silane pores, that is charge transfer at the pore bottom (a scarcely evident MF $\tau$, at $10$–$3 \times 10^{-2}$ Hz) and diffusion in the electrolyte through the pores (LF $\tau$, at $3 \times 10^{-2}$–$10^{-3}$ Hz). After 1 and 2 h, the LF $\tau$ was not detected, suggesting that at these short immersion times more impervious barrier characteristics were ensured.

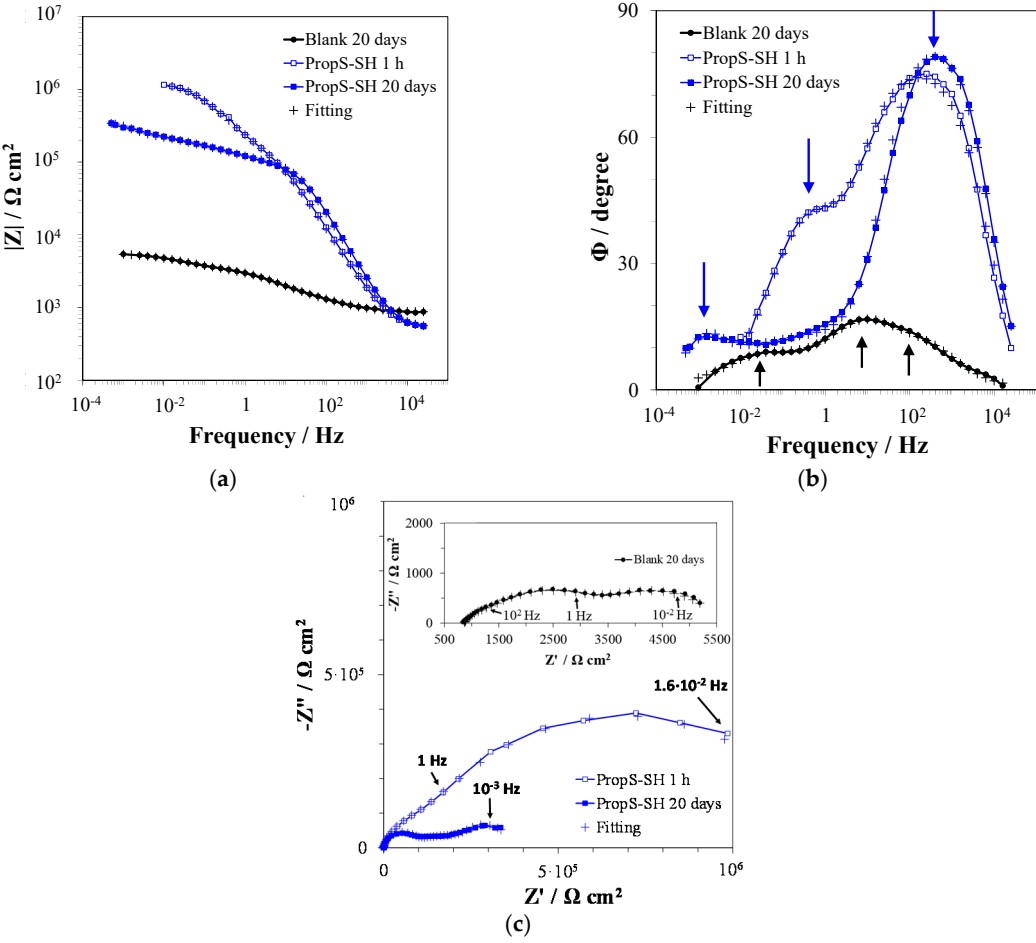

**Figure 7.** Bode (**a**,**b**) and Nyquist (**c**) plots recorded on bare and PropS-SH coated bronze in AR × 10 at 30 °C. Squares and circles represent experimental data and crosses are the result of the fitting.

Consequently, the ECs shown in Figure 8a,b were used to fit the EIS data at short and long immersion times, respectively. Both of them contain the following parameters:

- the solution resistance ($R_s$),
- the resistance and CPE of the silane film ($R_f$ and CPE$_f$), and
- the charge transfer resistance and the CPE of the double layer ($R_t$ and CPE$_{dl}$).

In addition, in Figure 8b a generalized finite length Warburg (GFLW) element was introduced, in series to the parallel $R_t$-CPE$_{dl}$ couple, with mathematical impedance expression:

$$Z_{GFLW} = R_W \frac{\tan h(j\omega T)^p}{(j\omega T)^p},$$

where $T$ is a time constant, $R_W$ is the polarization resistance of mass transport, and p is an exponent approaching but not higher than 0.5. For p = 0.5, $T = L^2/D$, where $L$ is the thickness of the diffusion layer and $D$ is the diffusion coefficient [47–49]. Other authors used the same or quite similar (containing a pure Warburg instead of a GFLW element) ECs to fit EIS spectra of silane coated copper [38,50,51], because these ECs correctly interpreted the physical condition of the coated metal.

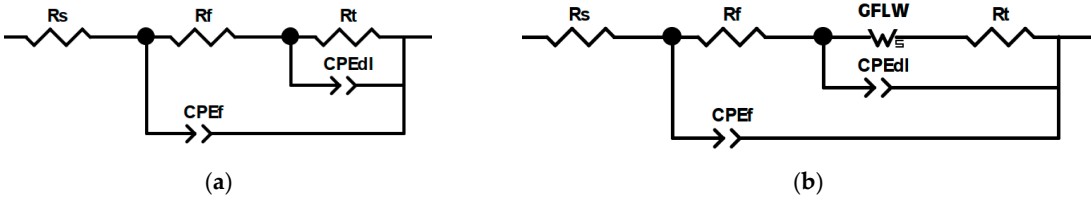

(**a**)            (**b**)

**Figure 8.** Equivalent circuits (EC) with two (**a**) or three (**b**) time constants used to fit the experimental Electrochemical Impedance Spectroscopy (EIS) spectra on bronze, protected by PropS-SH coatings non-containing or containing CeO$_2$, La$_2$O$_3$, and FA in AR × 10 at 30 °C.

The capacitive loops of plain PropS-SH spectra were much larger at 1 h than after 20 days of immersion (Figure 7), confirming the marked coating protectiveness at short immersion times. However, the Z modulus at the lowest investigated frequency still remained almost two orders of magnitude higher than that of the blank, suggesting the persistence of good protective properties towards the substrate even during long exposures. Figure 7 also includes the fitting curves that quite closely overlap the experimental data, evidencing the good fitting quality.

The spectra recorded after 1 h and 20 days on particle-charged PropS-SH coatings and the corresponding fitting curves are shown in Figure 9. The spectra shapes as a function of time are quite similar to those exhibited by plain PropS-SH. On CeO$_2$- and FA-loaded PropS-SH coatings, like for PropS-SH, the capacitive loops of the EIS spectra at 1-h immersion were larger than those obtained after 20 days, indicating a decrease in the coating barrier properties. Only in the presence of La$_2$O$_3$, the impedance moduli at the lowest investigated frequency after 1-h and 20 days were quite close to each other, suggesting the persistence of excellent coating performances throughout the immersion period.

According with immersion time, the ECs of Figure 8a,b proved quite suitable for fitting EIS spectra also in the case of nano and microparticle-loaded coatings. Figure 10 shows the time evolution of the EIS fitting parameters obtained from spectra of bare and coated bronze. The data related to bare bronze were those reported in [39]. The Figure clearly evidences that the particle loading produced coatings with higher $R_f$ values, suggesting more effective barrier properties in comparison to plain PropS-SH. The lower corresponding $C_f$ values indicated the formation of thicker coatings, in agreement with the results achieved on similar systems, including PropS-SH coatings loaded by CeAlO$_3$ nanoparticles applied on Mg alloys [30] or other nanoparticle-charged silane coatings [11,52].

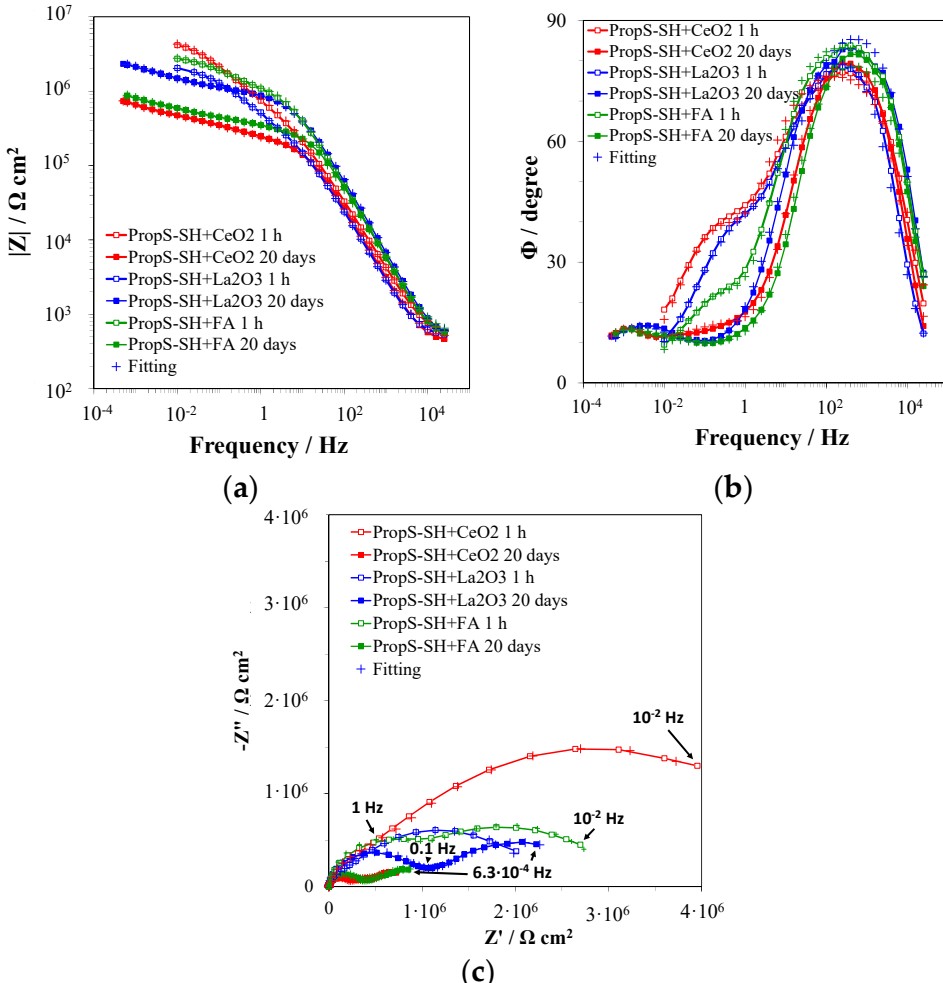

**Figure 9.** Bode (**a**,**b**) and Nyquist (**c**) plots recorded on bronze protected by PropS-SH coatings containing $CeO_2$, $La_2O_3$ and FA in ARx10 at 30 °C. Squares and circles represent experimental data and crosses are the result of the fitting.

The thickness ($d$) of the coatings can be estimated from the capacitance values at 1-h immersion by the relationship $d = \varepsilon_0 \varepsilon_r / C_f$ ($\varepsilon_0$ is the vacuum permittivity ($8.85 \cdot 10^{-14}$ F/cm), $\varepsilon_r$ the silane coating dielectric constant and $C_f$ the coating capacitance), if the dielectric constant of the silane film is known. Its value is about 10 for dry silane films [36]), but tends to increase with the extent of water penetration [36,53]. If calculated on the basis of $C_f$ and d values of PropS-SH coating (d is about 300 nm by the dip coating method [4]), $\varepsilon_r$ assumes values of about 68 and allows to roughly estimate the other coating thicknesses: that is 730 and 800 nm for $CeO_2$- and $La_2O_3$-loaded coatings, respectively, and about 1.7 μm, in the presence of FA.

Figure 10 evidences that the particle-loaded coatings exhibited $R_t$ values only slightly higher than those on plain PropS-SH, except in the case of $La_2O_3$ nanoparticles which determined a more pronounced increase, associated to much lower $C_{dl}$ values. These results suggested that this type of nanoparticles contributed to form a denser silane network leaving only a small substrate/electrolyte interface area at the pore bottom, available for charge transfer. In the presence of both micro and nanoparticles, the $R_w$ values of the GFLW element were higher than those of the plain coating and progressively increased, particularly in the presence of $La_2O_3$. According to some authors [5,35,36], this is due to the gradual obstruction of the pores in the silane network by insoluble corrosion products. In this view, the time increase in $T$ values could be connected to the growth of the diffusion layer thickness inside the coatings, induced by the precipitation of the corrosion products. In general, the

best fitting $n_f$ values of $CPE_f$ were close to unity (range 0.87–0.96), $n_{dl}$ values of $CPE_{dl}$ were in the range 0.55–0.80, while p values of the GFLW element were 0.35–0.5.

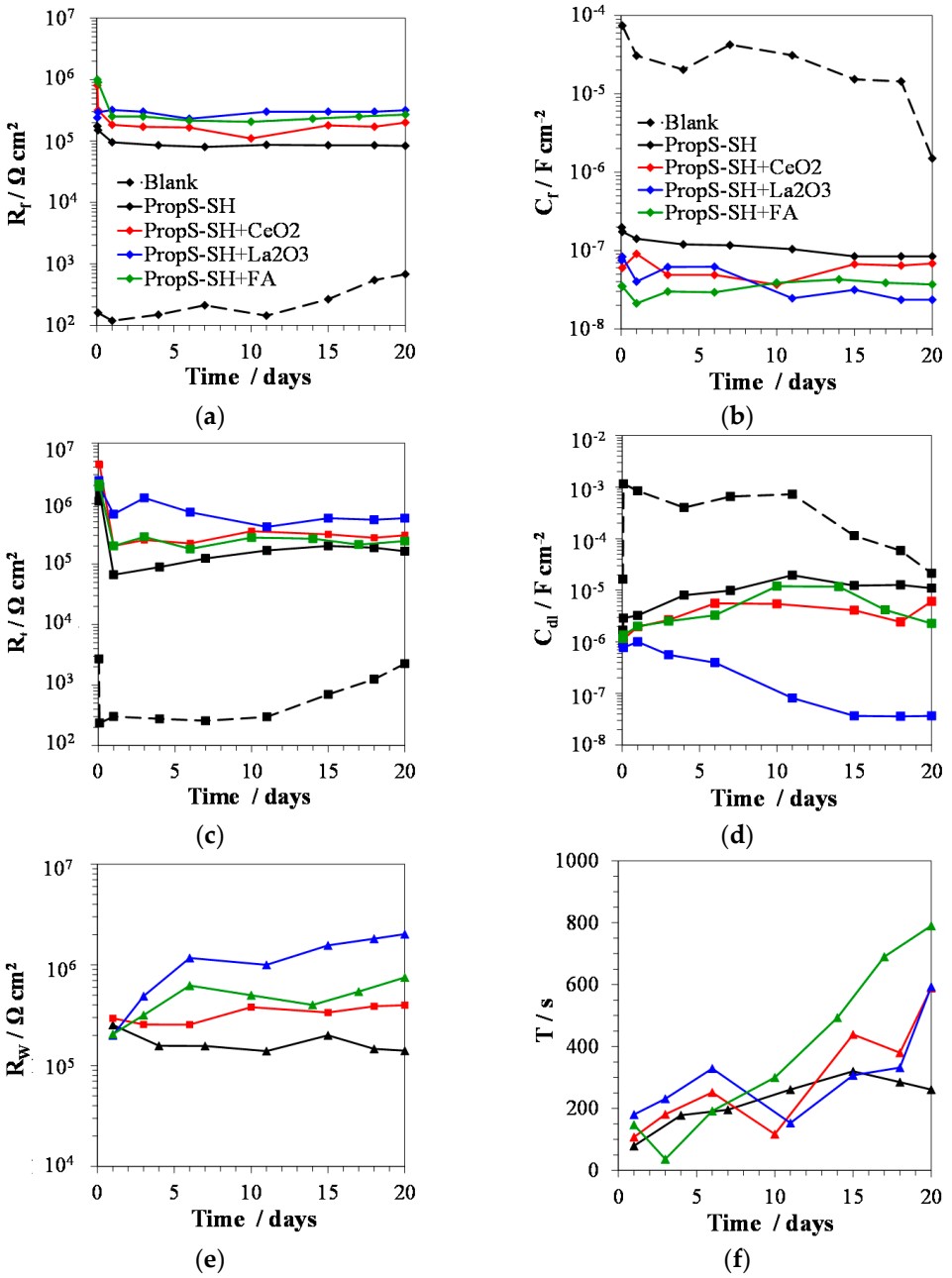

**Figure 10.** Evolution of the coating (or corrosion product film) resistance ($R_f$, **a**) and capacitance ($C_f$, **b**), charge transfer resistance ($R_t$, **c**) and double layer capacitance ($C_{dl}$, **d**), diffusion resistance ($R_W$, **e**), and diffusion time constant ($T$, **f**), during immersion of bare and coated bronze in AR × 10 at 30 °C. The coatings include PropS-SH and PropS-SH loaded with $CeO_2$, $La_2O_3$, and FA.

The polarization resistance ($R_p$) values for the electrolyte/coating/bronze systems represented by the ECs in Figure 8 are equal to the sum of $R_f$, $R_t$, and $R_W$ [48,54]. These data and the corresponding $E_{cor}$ values are collected in Figure 11, where the values related to bare bronze were again derived from [39]. In the blank, $R_p$ values were in the range 1–9 kΩ cm². In particular, relatively high values (8.6 kΩ cm²) were obtained at 1-h immersion, then $R_p$ decreased during the first day of immersion and remained rather constant at about 1 kΩ·cm² until about 10 days of immersion. Finally, the $R_p$ values increased to 5 kΩ·cm² at the end of the immersion period. Meanwhile, the corresponding $E_{cor}$ values

ennobled from $-0.10$ V(SCE) to about $0.00$ V(SCE) within 1 day and slowly evolved to $0.04$ V(SCE) after 20 days of immersion. In agreement with the polarization curve indications (Figure 4), PropS-SH coating induced much higher $R_p$ values, even if they decreased from 3 MΩ·cm$^2$ to 800 kΩ·cm$^2$ during the first immersion day, then slowly tended to 390 kΩ·cm$^2$ at the end of the immersion. In the presence of the nano- and microparticles, $R_p$ values in the range 3–5 MΩ·cm$^2$ were obtained after 1-h immersion. At the end of the immersion, $R_p$ decreased to 0.9–1.3 MΩ·cm$^2$ in the case of FA and CeO$_2$, respectively. Instead, in the presence of La$_2$O$_3$ $R_p$ values of 3 MΩ·cm$^2$ were finally achieved, quite similar to the initial ones. The $E_{cor}$ values of coated bronze were initially nobler than those achieved on bare bronze, then final values of 0.01–0.03 V(SCE) were reached both in the absence and in the presence of particle loading.

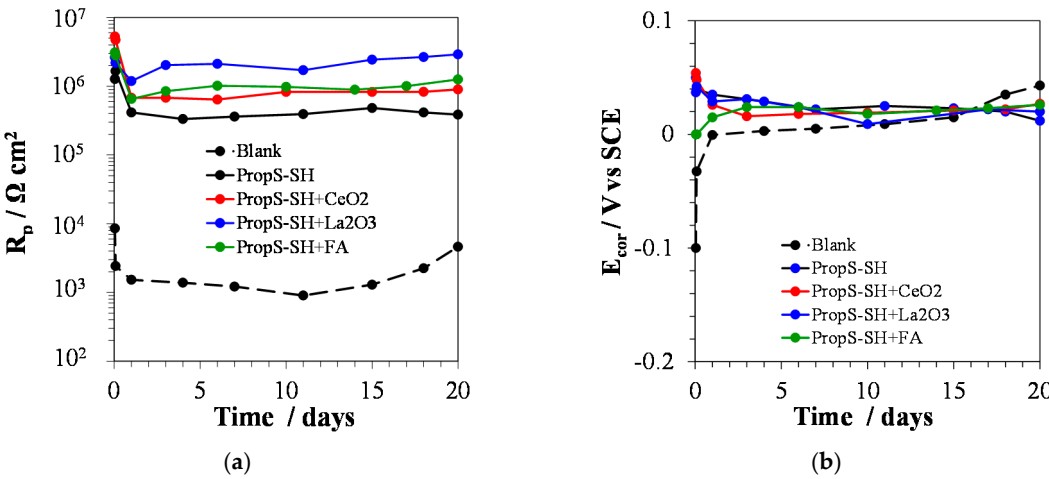

**Figure 11.** Evolution of polarization resistance ($R_p$) (**a**) and corrosion potential ($E_{cor}$) (**b**) values, during immersion of bare and coated bronze in AR × 10 at 30 °C. The coatings include PropS-SH and PropS-SH loaded with CeO$_2$, La$_2$O$_3$, and FA.

### 3.5. SEM-EDS Observations

The SEM micrographs in Figure 12 show the surface aspect of bare and coated specimens after 20 days of exposure to AR × 10. Unprotected bronze is covered by corrosion products, mainly cuprite and posniakite (Figure 12a), while only small amounts of corrosion products are observed on PropS-SH-prefilmed surfaces (Figure 12b), mainly in correspondence of bronze porosities. In the presence of CeO$_2$, La$_2$O$_3$, or FA nanoparticles (Figure 12c,d,f, respectively), no clear evidence of corrosion products was revealed. Typical EDS spectra of the dark round spots detected on La$_2$O$_3$- and FA- loaded PropS-SH coatings are shown in Figure 12e,g. In La$_2$O$_3$ coating, the spot composition is that of the coating itself, so that the spots are likely connected to denser coating regions, while in FA coating, the spots are Si and Al-rich zones and are related to the micrometric entrapped FA particles. In plain and CeO$_2$ loaded coatings, the few dark stains correspond to bronze shrinkage pores.

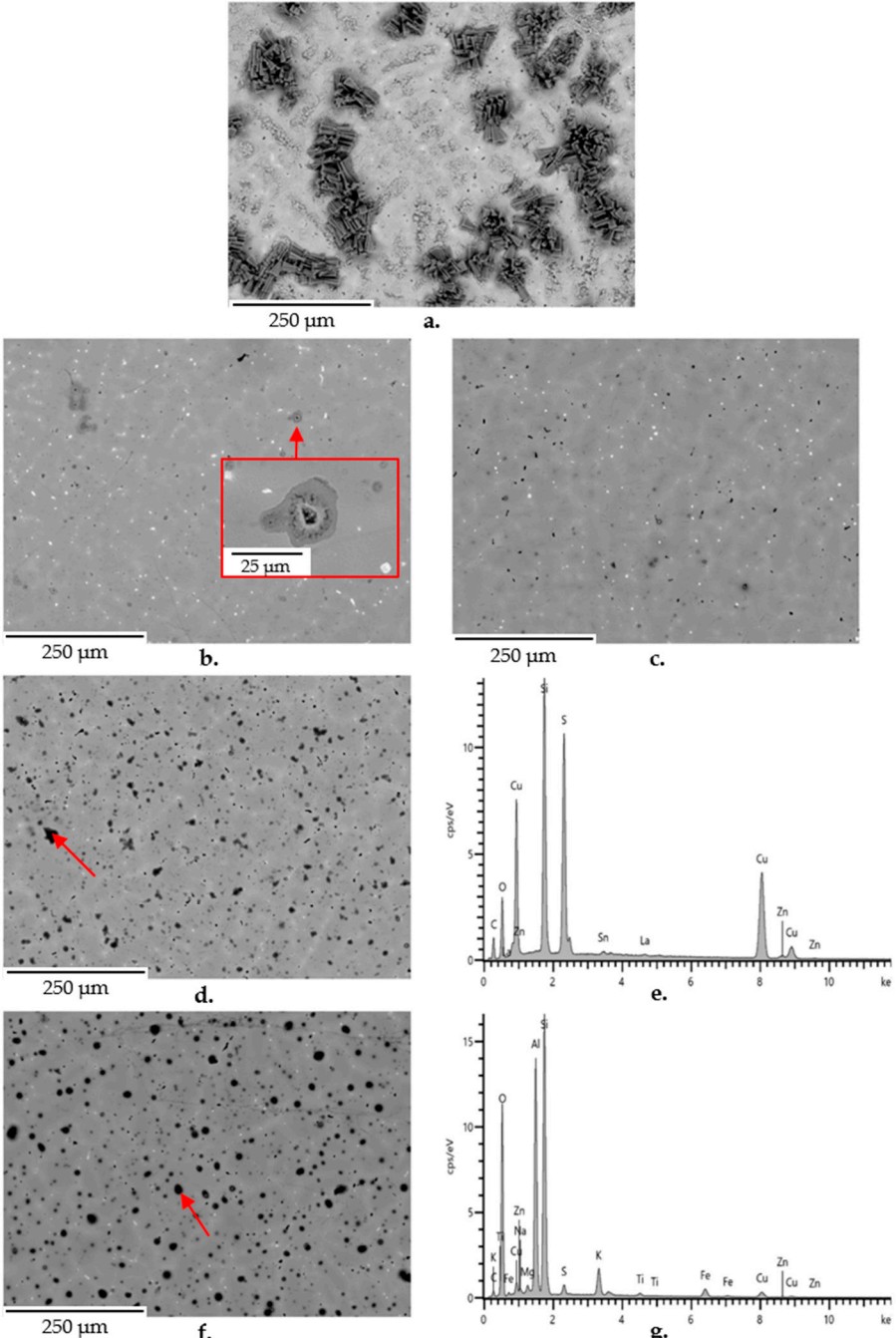

**Figure 12.** Back scattered electron SEM micrographs showing the surface aspects of bare (**a**) and coated (**b,c,d,f**) specimens after 20 days in AR × 10. The coatings include PropS-SH (**b**) and PropS-SH loaded with CeO$_2$ (**c**), La$_2$O$_3$ (**d**), and FA (**f**). The SEM- Energy Dispersion Spectroscopy (EDS) spectra in (**e**) and (**g**) refer to the elemental analysis of the black spots indicated by the arrows in Figure (**d**) and (**f**), respectively.

### 3.6. Tests in FA Suspensions

Figure 13 collects the $R_p$ and $E_{cor}$ values of bronze electrodes during immersions in AR × 10 in the absence and in the presence of 1 g/l FA. The R$_p$ data in FA suspensions were obtained by circle fitting the low frequency arc of the EIS spectra and by evaluating the limit of the real part of the impedance at frequency tending to zero. The figure shows that FA clearly hindered bronze corrosion, because

it determined $R_p$ values of two orders of magnitude higher and $E_{cor}$ values nobler than those in the blank solution.

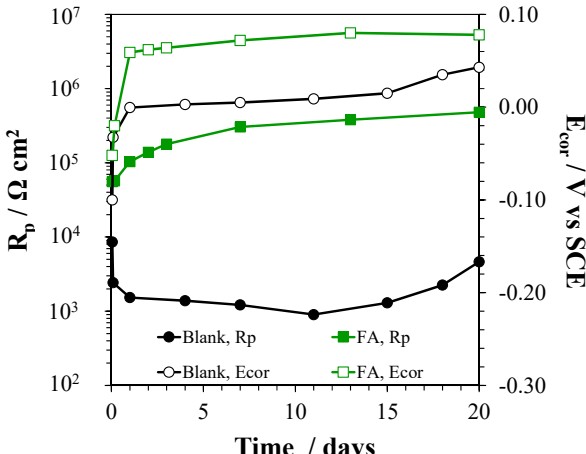

**Figure 13.** Time dependence of $R_p$ (solid symbols) and $E_{cor}$ (empty symbols) of bronze in AR × 10 both in the absence (dots) and in the presence (squares) of suspended FA particles.

Figure 14 gathers the polarization curves recorded at the end of the immersion period in the blank and in FA suspension. These curves agree with $R_p$ results as they show that FA slowed down both the cathodic and the anodic reactions of bronze corrosion and induced final $i_{cor}$ values of 0.1 μA·cm$^{-2}$, much lower than those in the blank solution (8.3 μA·cm$^{-2}$).

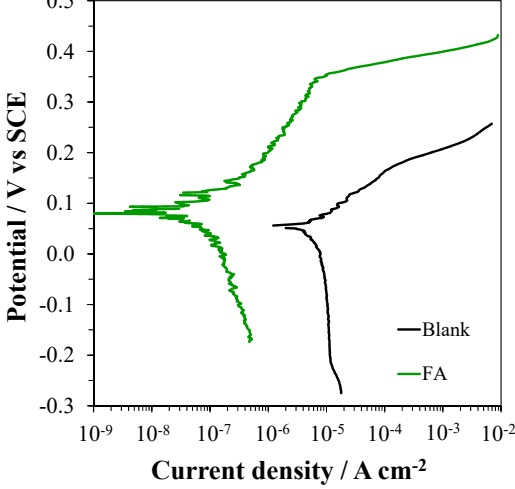

**Figure 14.** Polarization curves recorded on bare bronze after 20 days of immersion in AR × 10, in the absence (Blank) and in the presence (FA) of suspended FA particles.

Figure 15 shows the pH values recorded before and during the immersion. In the blank solution, during the electrode immersion a continuous pH increase from 3.3 to 5.6 was detected, due to the alkalization induced by oxygen reduction. Instead, a pH increase from 3.3 to 6.3 was detected in FA suspensions already during the aging period, suggesting the presence of soluble alkaline components in this additive, capable to mitigate the solution aggressiveness towards bronze corrosion. After the electrode immersion, the pH did not increase further, because of the significant suppression of bronze corrosion and oxygen reduction.

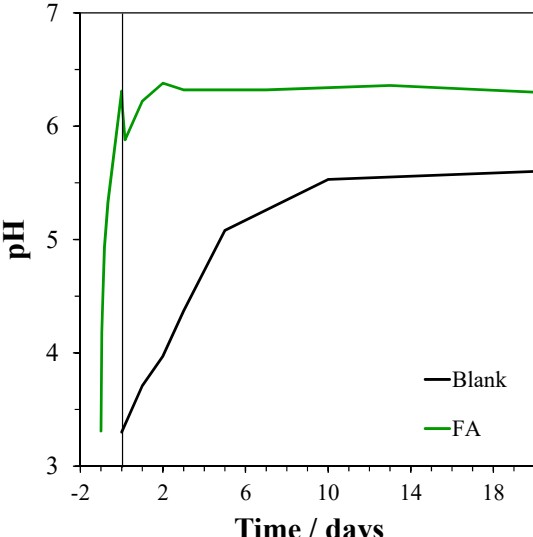

**Figure 15.** pH values of AR × 10 solution in the presence of suspended FA particles, recorded during the suspension aging and the subsequent immersion of bronze electrodes (starting at time = 0 days). As a reference, the solution pH of AR × 10 (blank) during immersion of bronze electrodes is also reported.

The high protectiveness offered by FA was also investigated by analyzing the concentrations of the elements leaching from FA particles into AR × 10, during 24 h stirring of the suspension. The results are collected in Table 5. Among the analyzed elements, Na was also present in AR × 10 formulation with the same concentration reported in Table 5. Therefore, no Na dissolution from suspended FA particles occurred, while the dissolved K amount was quite small. Instead, significant amounts of Ca and Mg were detected in the solution, likely due to the dissolution of small amounts of Ca and Mg oxides from FA and responsible of the pH neutralization of AR × 10 induced by FA addition. Therefore, the high $R_p$ values of bronze in FA suspensions were correlated to the increase in solution pH which mitigated the solution aggressiveness. Moreover, the presence of soluble Si species (about 0.7 mM) indicated the dissolution of silicates from FA, which can afford copper and copper alloys a moderate corrosion inhibition [55].

**Table 5.** Concentration of elements leaching from FA particles into AR × 10, during 24 h stirring of the suspension. The alkaline elements were analyzed by Atomic Absorption Spectroscopy (AAS), while the other elements were quantified by Inductively Coupled Plasma-Optical Emission Spectroscopy (ICP-OES).

| Element | Concentration/mg·L$^{-1}$ |
|---------|---------------------------|
| Al | 0.14 |
| Si | 19.1 |
| Ca | 98.9 |
| Fe | <0.32 |
| Mg | 18.9 |
| K | 0.27 |
| Na | 5.05 |

## 4. Discussion

The research here described evidenced the influence of the protocol adopted in silane coating preparation on the coating performances. Factors, such as the storage time at 4 °C, the presence of suspended micro and nanoparticles, the aging of nanoparticle suspensions, and the RT curing time were considered.

The tests showed that the coating performances improved at increasing storage times up to at least 1 year, indicating the prevailing of silane hydrolysis over silanol condensation throughout the storage period at 4 °C. This implies a shelf life of the hydrolyzed solution of at least 1 year, at this storage temperature. The coating performances also improved with the addition of FA, $CeO_2$, and $La_2O_3$, but, in the case of the nanoparticles, the improvement only occurred if a sufficient aging time of the nanoparticle suspensions was ensured before the coating preparation. This indicated that the development of a specific interaction between hydrolyzed silane molecules and nanoparticles was essential for the improvement of the coating performances. This interaction could be related to the adsorption of hydrolyzed silane molecules on the nanoparticles and formation of metallo-siloxane bonds, such as Ce–O–Si or La–O–Si bonds, through the condensation of silanols and hydroxyl groups on the nanoparticle surface. This phenomenon is expected to stimulate the silane network reticulation [15,24]. In contrast, no improvement in silane coating protectiveness was achieved by $TiO_2$ addition. The reason for this different behavior is still unclear, but could be connected to differences in the surface charging state of the nanoparticles which may hinder (in the case of $TiO_2$) or favor (for $CeO_2$ and $La_2O_3$) the surface bond development [23].

Finally, it was found that RT curing was possible and that the coating protectiveness increased at longer RT curing time up to at least 10 days, so complying with cultural heritage requirements for the protection of bronze artworks.

The EIS tests gave more detailed information about the influence of nano- and microparticles on the coating barrier properties. They evidenced the buildup of thicker coatings both in the presence of $CeO_2$ and $La_2O_3$ nanoparticles, likely due to surface silane adsorption and easier reticulation, and particularly after FA microparticle addition. In the latter case, FA is expected to favor the growth of thicker silane coatings, due to the alkalization effect they provoke (Figure 15). The pH increase in the hydrolyzed silane solution at least in proximity of the particle themselves is expected to stimulate reticulation and fast coating buildup because silanol condensation is reported to be alkaline-aided [23].

Both unloaded and nanoparticle- or microparticle-loaded coatings were affected by the presence of interconnected pores. In fact, the bronze EIS spectra were well interpreted by a porous coating model [13,38,56]. With the exception of the $La_2O_3$-loaded coating, the silane surface films were characterized by an increase in electrical conductivity during the first day of immersion (likely due to a progressive coating permeation by the aggressive solution) and then by more or less constant conductivity values, as indicated by the initial decreasing and then rather stable $R_f$ values. The concomitant $R_t$ decrease observed during the first day indicated a parallel stimulation of charge transfer at the pore bottom. In the presence of $La_2O_3$, rather constant $R_f$ and $R_t$ values were achieved throughout the immersion period, indicating that this film afforded more constant protection towards the substrate. In all coating types, diffusion affected bronze corrosion rates at immersion periods of 1 day or longer and, consequently, a GFLW element was introduced in the EC to correctly fit EIS spectra. With particle loading, the $R_W$ parameter increased with time, reasonably due to clogging of the fine coating pores by insoluble corrosion products [5,35,36]. Instead, in the absence of particles, the silane network was more pervious to corrosion product debris, as indicated by rather constant or even decreasing $R_W$ values. As a whole, the highest $R_f$, $R_t$, and $R_W$ values were achieved in the presence of $La_2O_3$, indicating the formation of the densest least porous silane network, which induced quite slow charge transfer and diffusion processes. The reasons for the outstanding coating characteristics afforded by $La_2O_3$ cannot be explained on the basis of the tests here described and need further specific investigations concerning the surface reactivity of these particles with hydrolyzed silane molecules, in association with tests on possible specific corrosion inhibition effects [24].

The tests performed on bare bronze specimens exposed to FA suspensions and the analysis of the leachate from FA particles into AR × 10 suggested that FA could increase the coating protectiveness not only by favouring the build-up of thicker coatings, but also by a more specific action inside the coatings themselves. In fact, in contact with the electrolyte permeating through the pores of the silane coating, FA particles can produce a local alkalization and can release soluble silicates, so reducing the

electrolyte aggressiveness towards bronze substrate and favoring silane pore plugging by entrapped insoluble corrosion products.

## 5. Conclusions

This research showed that the PropS-SH coating performances improved at increasing storage times up to 1 year, suggesting a shelf life of the hydrolyzed silane solution of at least 1 year, under refrigerated conditions of 4 °C.

The barrier properties of PropS-SH coatings increased by addition of both $CeO_2$ and $La_2O_3$ nanoparticles and FA microparticles. In the case of nanoparticles, this was observed only if the nanoparticle suspensions in silane solution underwent a sufficient aging time before coating preparation. The best performances were achieved in the presence of $La_2O_3$, while no improvement in silane coating protectiveness was obtained by $TiO_2$ addition.

The coatings were successfully cured at RT, so complying with cultural heritage requirements for the protection of bronze artworks.

According to EIS tests, PropS-SH loading by $CeO_2$ and $La_2O_3$ nanoparticles and FA microparticles determined an increase in coating thickness, likely due to easier silane reticulation. In the case of $CeO_2$ and $La_2O_3$ nanoparticles, this was likely favored by the adsorption of silane molecules on the particle themselves via metallo-siloxane bonds, while in the case of FA, quicker silanol condensation was favored in the proximity of the microparticles due to increased solution alkalinity on the particle surface.

Inside the coatings, partial FA dissolution is expected to increase the pH of the acidic solution penetrating through the coating pores and to produce soluble silicates, so reducing the electrolyte aggressiveness towards bronze corrosion and favoring silane pore plugging by entrapped insoluble corrosion products.

**Author Contributions:** Conceptualization and Methodology C.M., A.B., and F.Z.; Investigation V.G., F.Z., and M.S.; Writing original draft C.M., A.B., and F.Z. All authors have read and agreed to the published version of the manuscript.

**Funding:** This research was carried out in the scope of the B-IMPACT project (project ID 1149) of the M-ERA.NET 2013 call, supported by the national funding organizations of the partners Slovenia (MIZS), Italy (MIUR), and France (RMP).

**Conflicts of Interest:** The authors declare no conflict of interest.

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
