# Peer review of "Improving the Protectiveness of 3-Mercaptopropyl-Trimethoxysilane Coatings on Bronze by Addition of Oxidic Nano- and Microparticles"

_coatings, doi:10.3390/coatings10030225_

Round 1

Reviewer 1 Report

The goal of the paper is novel and the whole papers is worth to be published. The paper is well-written and easily readable. Research is done appropriate way and results are clearly discussed. The conclusions and presented coatings are interesting from both scientific and engineering point of view. Congratulate well-done job. Wishes. 

I have no comments on the manuscript.  

Author Response

Reviewer #1

The goal of the paper is novel and the whole papers is worth to be published. The paper is well-written and easily readable. Research is done appropriate way and results are clearly discussed. The conclusions and presented coatings are interesting from both scientific and engineering point of view. Congratulate well-done job. Wishes.

I have no comments on the manuscript. 

We thank the reviewer.

Reviewer 2 Report

Report on the manuscript “Improving the protectiveness of (3-mercaptopropyl)- 3 trimethoxysilane coatings on bronze by addition of 4 oxidic nano- and microparticles”, C. Monticelli et al.

Comments:

The manuscript describes a study on the preparation and performance of coatings based on 3-mercaptopropyl-trimethoxysilane (PropS-SH) incorporating (CeO2, La2O3, TiO2) and fly ash (FA) microparticles for the protection of bronze artworks. Overall, this is a technically correct, interesting manuscript so that publication is recommended after consideration of several minor remarks:

1) The Abstract is too long, containing generalizations corresponding to the Introduction section.

2) Polarization curves in Figs. 4 and 5 indicate that the protective capacity of the coatings is significantly diminished after 20 days of exposition to the corroding media relative to the performance after 1 h of exposition. It would be logical to perform experiments at longer times to evaluate the effective duration of the protective effect of the coatings.

3) The sentence in page 7, lines 208-210: “The application of PropS-SH coatings induced an excellent inhibition of both cathodic and anodic reactions after 1 h of immersion, with icor of 0.015 μA cm-2 208 . However, at the end of the 20 day 209 immersion, PropS-SH only hindered the oxygen reduction, so allowing final icor values of 0.5 μA cm-2 210 (Table 3).” should be clarified. What are the features denoting the hindering of the oxygen reduction?

4) La2O3 nanoparticles present a behavior slightly different to the others. A moder detailed description of these differences should be made.

5) The bias potential applies in EIS experiments should be indicated.

Author Response

Reviewer #2

The manuscript describes a study on the preparation and performance of coatings based on 3-mercaptopropyl-trimethoxysilane (PropS-SH) incorporating (CeO2, La2O3, TiO2) and fly ash (FA) microparticles for the protection of bronze artworks. Overall, this is a technically correct, interesting manuscript so that publication is recommended after consideration of several minor remarks:

1) The Abstract is too long, containing generalizations corresponding to the Introduction section.

 The abstract has been substantially modified by deleting the initial introductory part and by adding the main results.

2) Polarization curves in Figs. 4 and 5 indicate that the protective capacity of the coatings is significantly diminished after 20 days of exposition to the corroding media relative to the performance after 1 h of exposition. It would be logical to perform experiments at longer times to evaluate the effective duration of the protective effect of the coatings.

The polarization curves in Figures 4 (for PropS-SH), 5 (for PropS-SH+FA) and 6 (for PropS-SH+La2O3 and PropS-SH+CeO2) show that a significant decrease in coating performances was mainly achieved for plain PropS-SH, while the performances were much more stable with nano and microparticles, particularly for PropS-SH+La2O3. Therefore, the objective of achieving a coating with rather constant performances was reached.

In addition, we wish to stress that the described research does not aim at verifying how long the protectiveness of these coatings persists. It is a comparative research carried out to evaluate the possibility of improving the performances of plain PropS-SH by nano and microparticle addition.

Assessing the corrosion behaviour of bronze protected by these coatings at very long immersion times is reputed relatively important, because coatings for corrosion protection of actual bronze artworks will be thicker (at least some µm) than these and will not be prepared by dip coatings, but by spraying or by brushing, in order to increase the coating barrier properties.

On the other hand, the study of the influence of nano and microparticles addition on thicker coatings (already characterized by high protectiveness) would have been disappointingly time-consuming.

3) The sentence in page 7, lines 208-210: “The application of PropS-SH coatings induced an excellent inhibition of both cathodic and anodic reactions after 1 h of immersion, with icor of 0.015 μA cm-2. However, at the end of the 20 day immersion, PropS-SH only hindered the oxygen reduction, so allowing final icor values of 0.5 μA cm-2 (Table 3).” should be clarified. What are the features denoting the hindering of the oxygen reduction?

As already indicated at p. 5, line 176 and at p. 6, line 204, the cathodic polarization curve is connected to oxygen reduction. Therefore, as Figure 4 shows that after 20 days of immersion the cathodic curves on PropS-SH-coated bronze are characterized by lower currents than those recorded on bare bronze (blank at 20 days), it can be argued that after 20 days the oxygen reduction is still hindered by PropS-SH coating.

4) La2O3 nanoparticles present a behavior slightly different to the others. A moder detailed description of these differences should be made.

As now more clearly indicated at p. 17 line 431 and line 461, the reasons for the outstanding coating characteristics afforded by La2O3 cannot be explained on the basis of the tests here described and need further specific investigations concerning the surface reactivity of these particles with hydrolyzed silane molecules, in association with tests on possible specific corrosion inhibition effects.

5) The bias potential applies in EIS experiments should be indicated.

As indicated at p. 4 line 142, for EIS tests the applied potential signal was ± 10 mV rms.

Reviewer 3 Report

The work presented in this paper is well designed, structured, written and presented. I cannot find anything that needs to be corrected except some small typing details:

  • Unify the nomenclature used for 3-mercaptopropyl-trimetoxysilane along the paper, some times is written with parenthesis, some without, other abbreviated...
  • Line 73 “coating performance” instead of “coating performances”
  • Line 102 “consisted of” instead of “consisted in”
  • Line 265 blue arrows in figure 7b

The following comments are only small suggestions:

Page 6, lines 195-196 (fig 4):

Though anodic current present a marked segment with clearly lower current density slope, I wouldn´t think it is representative of a passive condition.

The cathodic polarization curve at 1 day of immersion has a particular profile. Do you have any hypothesis about it?

Page 9 EIS

It would be useful to compare with other studies to include a table with EIS fitting results (e.g. as supplementary information).

Line 259-261 Again, though there is a significant increase of impedance, I don´t think that after 1h immersion the alloy exhibits a strictly passive behavior;

Lines 298-99 “The ECs of Figure 8 proved quite suitable for EIS spectra fitting also in the case of nano and microparticle-loaded coatings”. EC in figure 8a, 8b or both according with immersion time? I suppose the situation is the same as with the no-loaded coating, but it is not completely clear in the sentence.

And as a final suggestion for future work, though the chosen methodology is convenient for laboratory comparison of different coatings, especially when many different conditions are being compared, it would be interesting for future studies to perform UV-CON weathering tests as a closer simulation to natural weathering conditions for outdoor monuments and/or compare to traditional and well known coatings (such as Incralac).

Author Response

Reviewer #3

The work presented in this paper is well designed, structured, written and presented. I cannot find anything that needs to be corrected except some small typing details:

  • Unify the nomenclature used for 3-mercaptopropyl-trimetoxysilane along the paper, some times is written with parenthesis, some without, other abbreviated...

  • Line 73 “coating performance” instead of “coating performances”
  • Line 102 “consisted of” instead of “consisted in”
  • Line 265 blue arrows in figure 7b

These corrections were introduced and highlighted in yellow in the text.

The following comments are only small suggestions:

Page 6, lines 195-196 (fig 4):

Though anodic current present a marked segment with clearly lower current density slope, I wouldn´t think it is representative of a passive condition.

The catalytic activity of bronze surfaces towards oxygen reduction depends on the presence or absence of surface oxides (like on copper [M.V. Vazquez, S.R. de Sanchez, E.J. Calvo, D.J. Schiffrin, The electrochemical reduction of oxygen on polycrystalline copper in borax buffer, Journal of Electroanalytical Chemistry, 374 (1994) 189-197]) and on their compactness. In the case of Figure 4, the low cathodic current on bronze after 1h immersion induced a more active Ecor which in turn allowed investigating a wider portion of the anodic polarization curve of bronze. As now indicated at p. 6 line 201, the shape of the anodic curve was typical of passive or pseudopassive (taking into consideration the not negligible values of the passive currents, in the range 1-10 μA cm-2) behaviour.

Concerning the presence and the influence of air formed protective films on copper and bronze, it is mentioned in many papers [G. Kear, B.D. Barker b, F.C. Walsh Electrochemical corrosion of unalloyed copper in chloride media––a critical review, Corrosion Science 46 (2004) 109–135.; LIAO Xiao-ning, CAO Fa-he, CHEN An-na, LIU Wen-juan, ZHANG Jian-qing, CAO Chu-nan, In-situ investigation of atmospheric corrosion behavior of bronze under thin electrolyte layers using electrochemical technique,

Trans. Nonferrous Met. Soc. China 22(2012) 1239-1249.; K.M. Zohdy, M.M. Sadawy, M. Ghanem, Corrosion behavior of leaded-bronze alloys in sea water, Materials Chemistry and Physics 147 (2014) 878-883.] which confirmed that it determines less noble Ecor.

The cathodic polarization curve at 1 day of immersion has a particular profile. Do you have any hypothesis about it?

It is not unusual finding peaks overlapped to the usual shape of the cathodic polarization curves on bronze

[C. Deslouis, B. Tribollet, G. Mengoli, M.M. Musiani, Electrochemical behaviour of copper in neutral aerated chloride solution. I. Steady-state investigation, Journal of Applied Electrochemistry 18 (1988) 374-383.]. In NaCl solution, a peak around -0.5 VSCE is attributed to the reduction of surface CuCl, while a peak at about -0.9 VSCE is connected to the reduction of Cu2O. Their occurrence depends on the chloride concentration and the immersion time.

In the case of ARx10, a reduction peak around -0.03 VSCE was observed. ARx10 has a complex composition which is reported to induce on bronze the surface precipitation of cuprite and basic copper sulphates (like posnjakite [A. Balbo, C. Chiavari, C. Martini, C. Monticelli, Effectiveness  of  corrosion  inhibitor  films  for  the  conservation  of  bronzes  and  gilded bronzes, Corrosion Science 59 (2012) 204-212.]). Given the peak position (much nobler than those attributed to Cu2O reduction), the peak could be tentatively ascribed to the reduction of basic copper sulphate, but deeper specific studies are needed to investigate this point.

Page 9 EIS

It would be useful to compare with other studies to include a table with EIS fitting results (e.g. as supplementary information).

It is quite difficult to compare the EIS data, because of the great variety of systems analysed.

The dielectric information about the coatings depend on:

- their nature/hydrolysis time/ thickness/curing time/solution composition/time of immersion/temperature of each step

- presence/amount/nature/pre-aging time of particles.

Even the substrate may affect the coating characteristics by facilitating or hindering bond formation with the coatings which in turn affects the coating porosity.

Of course, even the corrosion reaction at the pore bottom  depends on the substrate nature/composition of the electrolyte/time of immersion/coating porosity.

For all these reasons Tables of comparison of the EIS parameters is quite difficult, if not impossible.

It is instead possible to compare  the ECs used to fit the data which are connected to the hypothesized physical model and to verify that in a lot of cases the same ECs used in this manuscript were proposed. Now at p. 10 line 288 we added a sentence with proper literature references.

Line 259-261 Again, though there is a significant increase of impedance, I don´t think that after 1h immersion the alloy exhibits a strictly passive behavior;

This point was discussed in reply to a previous question. In p. 9, line 266, the term passive was accompanied with “pseudopassive” on the basis of the values of the passive currents.

Lines 298-99 “The ECs of Figure 8 proved quite suitable for EIS spectra fitting also in the case of nano and microparticle-loaded coatings”. EC in figure 8a, 8b or both according with immersion time? I suppose the situation is the same as with the no-loaded coating, but it is not completely clear in the sentence.

At p. 10 line 307, we have modified the sentence according to this suggestion.

And as a final suggestion for future work, though the chosen methodology is convenient for laboratory comparison of different coatings, especially when many different conditions are being compared, it would be interesting for future studies to perform UV-CON weathering tests as a closer simulation to natural weathering conditions for outdoor monuments and/or compare to traditional and well known coatings (such as Incralac).

Thank you for your kind suggestion.

Reviewer 4 Report

The present work has attempted to improve the protective ability of (3-mercaptopropyl)-trimethoxysilane coatings on bronze by addition of oxide nanoparticles (CeO2, La2O3, TiO2) and fly ash (FA) microparticles. In my opinion, this work is of great significance and value, and the experimental results are sound and adequate. However, I also found some problems about the electrochemical results. The paper can be accepted after addressing all the issues.
1. The abstract should state the impact and vital results of this work, the present is a bit long, please improve it.
2. The introduction section is sound and well written.
3. Table 1, please verify the composition unit.
4. Please improve the resolution of the Fig. 1, and add some amplified images.
5. Section 2.2 for EIS test, why you choose the frequency range from 25 kHz to 1 mHz, rather than 100kHz-10mHz?
6. Results, Figs.2-4, the unit of potential (Vsce) is wrong, it should be V vs. SCE?
7. As for the EIS data, Figs. 7a-b, the unit of log f is wrong.
8. For EIS fitting, Fig. 8, the equivalent circuits b shows three time constants, including a Warburg resistance, however, according to Fig. 7-Nyquist plot, no diffusion process can be found, why do you use Warburg resistance to fit the data? Please verify it and provide the new fitted results.

Author Response

Reviewer #4

The present work has attempted to improve the protective ability of (3-mercaptopropyl)-trimethoxysilane coatings on bronze by addition of oxide nanoparticles (CeO2, La2O3, TiO2) and fly ash (FA) microparticles. In my opinion, this work is of great significance and value, and the experimental results are sound and adequate. However, I also found some problems about the electrochemical results. The paper can be accepted after addressing all the issues.

  1. The abstract should state the impact and vital results of this work, the present is a bit long, please improve it.

The abstract has been substantially modified by deleting the initial introductory part and by adding the main results.

  1. The introduction section is sound and well written.

Thank you

  1. Table 1, please verify the composition unit.

The composition units are correct but the table borders have been modified in order to clarify that the measure units of the Concentrations, Electrical conductivity and pH are obviously different. For homogeneity, also the format of the other Tables has been modified accordingly.

  1. Please improve the resolution of the Fig. 1, and add some amplified images.

Figure 1 has been splitted in Figure 1a and 1b in order to show FA particles at two different magnifications. The high resolution files of these Figures have been uploaded with the revised version of the manuscript.

  1. Section 2.2 for EIS test, why you choose the frequency range from 25 kHz to 1 mHz, rather than 100kHz-10mHz?

The very high frequency range was neglected in recording EIS spectra because it often exhibits one (or more) loops  irrelevant to the electrode process, connected to the impedance of the reference electrode, cables and wires [S. Chechirlian, M. Keddam, and H. Takenouti, "Specific Aspects of Impedance Measurements in Low Conductivity Media," in Electrochemical Impedance: Analysis and Interpretation, ed. J. Scully, D. Silverman, and M. Kendig (West Conshohocken, PA: ASTM International, 1993), 23-36. https://doi.org/10.1520/STP18061S ; https://www.biologic.net/documents/eis-precautions-electrochemistry-battery-application-note-5/].

  1. Results, Figs.2-4, the unit of potential (Vsce) is wrong, it should be V vs. SCE?

The titles of the Figures have been changed to V vs SCE, while in the text for the sake of simplicity VSCE has been substituted by V(SCE).

  1. As for the EIS data, Figs. 7a-b, the unit of log f is wrong.

The titles of Figs. 7a,b, 9a,b and have been changed into Frequency / Hz and  ½Z½/ Ω cm2.

  1. For EIS fitting, Fig. 8, the equivalent circuits b shows three time constants, including a Warburg resistance, however, according to Fig. 7-Nyquist plot, no diffusion process can be found, why do you use Warburg resistance to fit the data? Please verify it and provide the new fitted results.

The analysed EIS spectra included three time constants and were properly fitted by the ECs of Figure 8. This is clearly demonstrated by the overlapping of the experimental spectra in Figures 7 and 9 and the corresponding fitting spectra. These ECs agree with a physical model of a metal substrate protected by porous coatings, where the substrate corrosion depends on both the coating barrier properties and charge transfer (EC of Figure 8a) or charge transfer / diffusion (EC of Figure 8b).  These ECs were  already used by other authors [38,50,51 in the manuscript].

The Warburg element adopted (Generalized Finite Length Warburg, Ws) terminates in a finite resistance (see ZView®, IMPEDANCE / GAIN PHASE  GRAPHING AND ANALYSIS SOFTWARE, OPERATING MANUAL, Version 3.5 and Figure below]. The presence of diffusion is not always evident like in the reference case of the Figure below (obtained with Ws-R = 1000; Ws-T = 0.1; Ws-P = 0.5),  because it depends on how Ws couples to the other circuit elements.

[Figure, see pdf document uploaded]

Figure Nyquist plot of GFLW obtained with Ws-R = 1000; Ws-T = 0.1; Ws-P = 0.5 [ZView®, IMPEDANCE / GAIN PHASE  GRAPHING AND ANALYSIS SOFTWARE, OPERATING MANUAL, Version 3.5].
